# Multilayer Epitaxial Heterostructures with Multi-Component III–V:Fe Magnetic Semiconductors

**DOI:** 10.3390/nano13172435

**Published:** 2023-08-28

**Authors:** Alexey V. Kudrin, Valeri P. Lesnikov, Ruslan N. Kriukov, Yuri A. Danilov, Mikhail V. Dorokhin, Anastasia A. Yakovleva, Nataliya Yu. Tabachkova, Nikolai A. Sobolev

**Affiliations:** 1Research Institute for Physics and Technology, Lobachevsky State University of Nizhny Novgorod, Gagarin av. 23/3, 603950 Nizhny Novgorod, Russia; kudrin@nifti.unn.ru (A.V.K.); valeri.lesnik@yandex.ru (V.P.L.); kriukov.ruslan@yandex.ru (R.N.K.); danilov@nifti.unn.ru (Y.A.D.); dorokhin@nifti.unn.ru (M.V.D.); nastayakovleva42@gmail.com (A.A.Y.); 2Department of Materials Science of Semiconductors and Dielectrics, National University of Science and Technology “MISiS”, 119049 Moscow, Russia; ntabachkova@misis.ru; 3Prokhorov General Physics Institute, Russian Academy of Sciences, 38 Vavilov st., 119991 Moscow, Russia; 4Department of Physics and i3N, University of Aveiro, 3810-193 Aveiro, Portugal; 5Laboratory of Functional Low-Dimensional Structures, National University of Science and Technology “MISiS”, 119049 Moscow, Russia

**Keywords:** magnetic semiconductors, pulsed laser deposition, III–V heterostructures, transmission electron microscopy, X-ray photoelectron spectroscopy, magnetoresistance

## Abstract

Three-layer structures based on various multi-component films of III–V semiconductors heavily doped with Fe were grown using the pulsed laser sputtering of InSb, GaSb, InAs, GaAs and Fe solid targets. The structures comprising these InAsSb:Fe, InGaSb:Fe and InSb:Fe layers with Fe concentrations up to 24 at. % and separated by GaAs spacers were deposited on (001) *i*-GaAs substrates at 200 °C. Transmission electron microscopy showed that the structures have a rather high crystalline quality and do not contain secondary-phase inclusions. X-ray photoelectron spectroscopy investigations revealed a significant diffusion of Ga atoms from the GaAs regions into the InAsSb:Fe layers, which has led to the formation of an InGaAsSb:Fe compound with a Ga content up to 20 at. %. It has been found that the ferromagnetic properties of the InAsSb:Fe magnetic semiconductor improve with an increasing Sb:As ratio. It has been concluded that the indirect ferromagnetic exchange interaction between Fe atoms occurs predominantly via Sb atoms.

## 1. Introduction

Spin electronics (spintronics) is the field of science and technology based on the use, besides the charge, of another fundamental property of the electron: its spin. The global goal of semiconductor spintronics is the creation of a new elemental base of electronics operating on new physical principles. For semiconductor electronics, the use of the spin of charge carriers may open new perspectives after reaching the size limit of traditional approaches (about 3 nm). The solution to the practical problems of semiconductor spintronics depends on the creation of new functional semiconductor nanomaterials with controlled electrical, magnetic and optical properties. The presence of intrinsic ferromagnetic properties in semiconductors enables spin-dependent transport and optical effects that are inaccessible in nonmagnetic semiconductors. The main reproducible results in the field of the fabrication of magnetic semiconductors were achieved in III–V semiconductors doped with Mn, in particular, in structures based on GaAs heavily doped with Mn (with the Mn concentration up to ~10 at. % [1,2]). At the same time, the Curie temperature of (Ga,Mn)As does not exceed 200 K, which obviously is the upper limit for this material [1]. Regardless, the nature of the magnetism in (Ga,Mn)As remains a subject of discussion. Despite numerous experimental and theoretical works, the theories of magnetism in (Ga,Mn)As are divided into two classes: the theory of indirect exchange interaction via the ensemble of holes in the valence band (Zener’s *p*-*d* exchange model) and the theory of the double exchange via more localized carriers in the Mn impurity band (Zener’s double exchange model) [2]. The uncertainty in understanding the physics of the magnetism in (Ga,Mn)As makes it difficult to improve the properties of this material and develop devices based on it. Recent investigations in the field of semiconductor spintronics show that III–V semiconductors doped with iron instead of manganese are more promising. The active study of the III–V:Fe magnetic semiconductors began about a decade ago, when ferromagnetic (In,Fe)As layers with a Curie temperature up to 70 K at a Fe concentration of 4 at. % were obtained via molecular beam epitaxy (MBE) [3]. Subsequently, the epitaxial layers of the III–V semiconductors heavily doped with Fe, with a Curie point up to room temperature, were obtained. In particular, (Ga,Fe)Sb [4,5], (In,Fe)Sb [6,7], and GaAs:Fe [8] epitaxial layers with concentrations of Fe atoms up to 20 at. % and Curie temperatures above 300 K were obtained via MBE and/or pulsed laser deposition (PLD). Despite the very high Fe doping level, these layers are single-phase ferromagnetic semiconductors with the intrinsic mechanism of ferromagnetic ordering. Since III–V:Fe compounds are rather new magnetic semiconductors, the general theory of ferromagnetism in such materials is absent so far. In (In,Fe)As layers, the ferromagnetic exchange was interpreted within the theory of indirect exchange via conduction electrons [3]. At the same time, for the (Ga,Fe)Sb and (In,Fe)Sb layers it was shown that ferromagnetic exchange is not carrier-mediated [4,6]. For (Al,Fe)Sb insulating layers it was assumed that the ferromagnetic exchange is associated with some ferromagnetic super-exchange mechanism [9]. For the ferromagnetic GaAs:Fe layers, the indirect ferromagnetic exchange was interpreted in terms of Zener’s double exchange model [8]. Unlike magnetic III–V semiconductors doped with Mn, the III–V:Fe magnetic semiconductors can be either *n*-type ((In,Fe)As, (In,Fe)Sb, GaAs:Fe) or *p*-type (Ga,Fe)Sb), which is necessary for the implementation of bipolar spintronic devices. In particular, this makes it possible to create *n*-*p* diode structures with two III–V:Fe magnetic semiconductor layers of different conductivity types, such as (Ga,Fe)Sb and (In,Fe)Sb [10,11] as well as (Ga,Fe)Sb and (In,Fe)As [11]. It is also possible to form ferromagnetic layers based on heavily Fe-doped III–V multi-component semiconductor solid solutions. In particular, Fe-doped ferromagnetic layers based on an In_1-*x*_Ga*_x_*Sb matrix were obtained [12]. The interest in multi-component III–V:Fe magnetic semiconductor solid solutions stems from the unique possibilities of manipulating the basic semiconductor parameters (the type of majority charge carriers, the band gap and the crystal lattice parameters) by varying the composition of the III–V matrix. This will potentially expand the use of the III–V:Fe magnetic semiconductors towards the creation of semiconductor spintronics devices.

Thus, the III–V:Fe high-temperature ferromagnetic semiconductors are currently in the active phase of research, therefore, new achievements in the technology of their creation and the revelation of the new features of their magnetic and transport properties add new knowledge to the physics of magnetic semiconductors, as well as to the physics of semiconductors and condensed matter physics in general. In this work, we present the results of the fabrication of three-layer heterostructures with III–V:Fe magnetic semiconductor layers based on the In_1-*x*_Ga*_x_*Sb and InAs_1-*x*_Sb*_x_* solid solutions.

## 2. Materials and Methods

The structures presented in this work were grown on semi-insulating (001) GaAs substrates using the pulsed laser sputtering of InSb, GaSb, InAs, GaAs and Fe solid targets in a vacuum [5,6]. The nominal composition of the formed layers was set using the ratio of the target sputtering times. To form the InAsSb:Fe layers, the cyclic serial sputtering of the InAs, InSb and Fe targets was carried out. The InGaSb:Fe layers were formed using the InSb, GaSb and Fe targets. The temperature of the growth process for all structures was 200 °C since this temperature was earlier found to be optimal for the growth of III–V:Fe magnetic semiconductors by PLD [6,8,10]. The structural properties of the fabricated samples were investigated using high-resolution transmission electron microscopy (HRTEM). The composition of the structures was studied using X-ray photoelectron spectroscopy (XPS). The concentration of elements was determined using relative sensitivity factors [13]. The absolute error in the determination of the atom concentration in the used XPS method is 1 at. %. Magnetotransport measurements were carried out using van der Pauw geometry in a closed-cycle He cryostat.

## 3. Results

### 3.1. Three-Layer Structures

Three types of three-layer structures containing two III–V:Fe layers separated by a GaAs spacer were formed and studied:

Sample M1—an InAs_0.5_Sb_0.5_:Fe and an InAs_0.8_Sb_0.2_:Fe layer separated by an undoped GaAs layer. The nominal Fe concentration was set equal to 17 at. %.

Sample M2—an InAs_0.5_Sb_0.5_:Fe and an In_0.2_Ga_0.8_Sb:Fe layer separated by an undoped GaAs layer. The nominal Fe concentration was set equal to 17 at. % and 20 at. % for the InAs_0.5_Sb_0.5_:Fe and In_0.2_Ga_0.8_Sb:Fe layers, respectively.

Sample M3—an In_0.2_Ga_0.8_Sb:Fe and an InSb:Fe layer separated by an undoped GaAs layer. The nominal Fe concentration was set equal to 20 at. %.

The nominal thickness of each layer was set equal to 25 nm. The structures are listed in Table 1.

Figure 1a shows an overview cross-section TEM image and Figure 1b shows an HRTEM image of sample M1. The images clearly reveal all grown layers: the lower (closest to the substrate) InAsSb:Fe layer (ca. 33 nm thick), the GaAs spacer layer (ca. 29 nm thick) and the upper InAsSb:Fe layer (ca. 25 nm thick). All three deposited layers are flat and epitaxial. No secondary-phase inclusions were observed (they may be visualized in III–V:Fe layers in the form of regions with a moiré-type contrast [6]). The images reveal the presence of microtwins which appear in the form of straight lines at an angle of ca. 70° with respect to each other. The microtwins are typical crystal defects for the formed layers due to the large lattice mismatch between the InAsSb and GaAs crystals. The inset of Figure 1b shows a fast Fourier transform (FFT) diffraction pattern of the whole HRTEM image. The pattern contains spots corresponding to a zinc-blende-type crystal structure for the GaAs (substrate and intermediate layer) and InAsSb:Fe regions. The diffraction spots located at a smaller distance from the central spot in the reciprocal space (the inner spots denoted as MS—magnetic semiconductor) correspond to the InAsSb:Fe layers with a larger lattice parameter, while the spots with a larger distance correspond to the GaAs regions with a smaller lattice parameter.

Figure 1c,d exhibit inverse FFT images obtained from the MS and GaAs spots of the FFT diffraction pattern. The images clearly show that the GaAs regions and the InAsSb:Fe layers have different lattice parameters, and the lattice parameter of the GaAs spacer is close to that of the GaAs substrate. It should be noted that the change in the lattice parameter (layer relaxation) occurs in a very thin region (a few nanometers thick). The analysis of the HRTEM image shows that the lattice parameter of the lower InAsSb:Fe layer is larger than that of the GaAs spacer by ca. 13 %. At the same time, the lattice parameter of the upper InAsSb:Fe layer is larger than that of the GaAs spacer by ca. 8 %. This result agrees with the nominal composition of the InAsSb:Fe layers (Table 1), since the lower InAsSb:Fe layer contains a higher concentration of Sb in the InAsSb semiconductor matrix than the upper InAsSb:Fe layer. Note that the lattice mismatch between the GaAs and InSb amounts to 14.6 % and between the GaAs and InAs to 7.2 %.

Figure 2 shows cross-section HRTEM images of sample M1 obtained at a higher magnification in the region of the lower InAsSb:Fe layer (a), in the region of the GaAs spacer (b) and in the region of the upper InAsSb:Fe layer (c). It can be seen from the HRTEM images that all layers possess a fairly high crystalline quality and do not contain inclusions with a crystal lattice different from the zinc-blende type.

Figure 3 exhibits the XPS dependences of the concentration of C, O, In, Ga, As, Sb and Fe atoms on the distance from the surface for sample M1. The Fe concentration in the InAsSb:Fe layers detected using the XPS equals about 13–14 at. %, which is consistent with the nominal Fe concentration (17 at. %). The lower InAsSb:Fe layer has an Sb concentration approximately three times higher than that of the upper InAsSb:Fe layer, which is also in agreement with the technological parameters of the layers (Table 1). It is noteworthy that the layers with the nominal composition of InAsSb:Fe contain a significant concentration of Ga atoms (no less than 12 at. % in the lower layer and no less than 4 at. % in the upper layer) although Ga was not introduced during the growth of these layers. The presence of Ga in these layers is apparently related to Ga diffusion from the GaAs substrate and spacer during the growth process. Actually, the formed magnetic semiconductor films are layers of the InGaAsSb quaternary solid solution heavily doped with Fe. It is also seen from the XPS profiles that Fe atoms tend to diffuse towards the direction opposite to the growth direction (towards the substrate) despite the relatively low temperature of the growth process (200 °C). The XPS data also reveal that, at a very high doping level, Fe atoms replace not only III elements but also partly V elements. This is the reason for the use of the III–V:Fe notation instead of (III,Fe)V. The XPS analysis of chemical bonds [8] reveals for sample M1 (as for M2 and M3) the absence of the Fe-Fe chemical bonds and isolated Fe atoms (within the error of the technique of about 1 at. %). This result indicates that the III–V:Fe layers for all our structures do not contain metallic iron clusters and that all introduced Fe atoms are in substitutional positions in the III–V lattice.

Figure 4a,b presents overview TEM and HRTEM images of sample M2. All layers of sample M2 can be revealed in the images: the lower InAsSb:Fe layer (ca. 23 nm thick), the GaAs spacer (ca. 20 nm thick) and the upper InGaSb:Fe layer (ca. 23 nm thick). The FFT diffraction pattern (inset to Figure 4b) contains spots related to the GaAs regions (the substrate and the spacer layer) and to the InAsSb:Fe/InGaSb:Fe magnetic semiconductor layers. The InAsSb:Fe and InGaSb:Fe regions give spots that do not differ in the FFT pattern due to the close lattice parameters of these layers.

Figure 5 shows the HRTEM images at a higher magnification in the region of the lower InAsSb:Fe layer (a), in the region of the GaAs spacer (b) and in the region of the upper InGaSb:Fe layer (c). All layers are epitaxial, single-phase and contain microtwins. From the HRTEM images, it was found that the lattice parameters of the InAsSb:Fe layer and the InGaSb:Fe layer differ from that of the GaAs spacer by about 10 % and 8 %, respectively (the lattice mismatch between the GaAs and GaSb single crystals equals 8 %).

Figure 6 displays the XPS depth profiles of the constituent elements for sample M2. The concentration of Fe atoms in the InAsSb:Fe layer (ca. 17 at. %) and the InGaSb:Fe layer (ca. 23 at. %) is consistent with the nominal Fe content for these layers (Table 1).

As for sample M1, the significant gallium diffusion from the substrate and spacer into the InAsSb:Fe layer occurs in sample M2. A consequence of this is the observation of about 20 at. % Ga in the InAsSb:Fe layer. There is also a significant diffusion of Fe atoms in the direction opposite to the growth direction, which leads to the appearance of a significant Fe concentration (ca. 10 at. %) in the GaAs spacer and GaAs substrate.

Figure 7 shows overview and high-resolution cross-section TEM images of sample M3. As for structures M1 and M2, all deposited layers are flat, epitaxial and uniform. The thickness of the lower InGaSb:Fe layer equals ca. 30 nm, that of the GaAs spacer layer is ca. 21 nm and that of the upper InSb:Fe layer is ca. 33 nm. The inset of Figure 7b shows the FFT diffraction pattern of the whole HRTEM. As for structures M1 and M2, the diffraction pattern contains the main reflections related to the GaAs regions (substrate and spacer) and the MS layers (InGaSb:Fe and InSb:Fe that have close lattice parameters). Also, the diffraction pattern contains a set of additional spots (denoted as MT) related to microtwins. Figure 7c,d show inverse FFT images obtained from the main reflexes while Figure 7e shows an inverse FFT image obtained from the additional reflexes. Figure 7e confirms that the additional diffraction spots correspond to the regions of microtwins that arise in the InGaSb:Fe layer at the boundary with the substrate and grow through all layers to the surface.

As for samples M1 (Figure 2) and M2 (Figure 5), all layers of sample M3 have quite a high crystalline quality, do not contain secondary-phase inclusions, and the main defects of the crystal structure are microtwins (Figure 8). From the HRTEM images, it was found that the lattice constants of the InGaSb:Fe layer and the InSb:Fe layers differ from the GaAs spacer lattice constant by about 9 % and 13 %, respectively.

Figure 9 shows the XPS dependences of the constituent elements’ concentrations on the distance from the surface for sample M3. The InGaSb:Fe and InSb:Fe layers have a very high Fe concentration (about 20–24 at. %), but despite this, the Fe-doped layers have a rather high crystalline perfection (Figure 8). As for structures M1 (Figure 3) and M2 (Figure 6), a significant Fe diffusion into the GaAs spacer and substrate from the Fe-doped layers was revealed for sample M3.

### 3.2. InAsSb:Fe/GaAs and InGaSb:Fe/GaAs Structures

Samples M1, M2, and M3, considered above, contain layers of III–V solid solutions heavily doped with Fe. According to the HRTEM and XPS data, these multi-component layers are single-phase with a Fe concentration in the range of 13–24 at. %. It is of interest to determine the basic transport and magnetotransport properties of the individual layers of such magnetic semiconductors. For this, the following single InAsSb:Fe and InGaSb:Fe layers with a nominal thickness of 35 nm were formed on *i*-GaAs (001) substrates at 200 °C:

Sample S1—a single InAs_0.8_Sb_0.2_:Fe layer with a nominal Fe content of 17 at. %.

Sample S2—a single InAs_0.5_Sb_0.5_:Fe layer with a nominal Fe content of 17 at. %.

Sample S3—a single In_0.2_Ga_0.8_Sb:Fe layer with a nominal Fe content of 20 at. %.

The technological (nominal) composition of the layers was set to be the same as in samples M1, M2 and M3. The structures are listed in Table 2.

Figure 10 presents the temperature dependences of the resistivity *ρ*(*T*) for samples S1 and S2. Sample S1, with a technological ratio of the Sb and As concentrations equal to 1:4, has a very weak *ρ*(*T*) dependence which is similar to the temperature dependences of the resistivity for MBE-grown (In,Fe)As layers [3]. The *ρ*(*T*) dependence for sample S2, with a nominal ratio of the Sb and As concentrations equal to 1:1, is much more pronounced and is similar to the *ρ*(*T*) dependences for the PLD-grown InSb:Fe layers [6].

Figure 11 shows Hall constant dependences on an external magnetic field, *R*_H_(*B*), in the temperature range of 30–130 K for sample S1. For all temperatures, the main contribution to the *R*_H_(*B*) dependence comes from the ordinary Hall effect corresponding to the *n*-type conductivity. The concentration and mobility of carriers (electrons) at 300 K obtained from the Hall effect data are about 1 × 10^19^ cm^−3^ and 40 cm^2^/(V·s), respectively. The carrier concentration in the obtained InAsSb:Fe layers is determined by the native point defects of the semiconductor matrix, which are predominantly donors in the InAs and InSb semiconductors. At temperatures below 70 K, the *R*_H_(*B*) dependences also have a component related to the anomalous Hall effect (AHE). At 10 K, the *R*_H_(*B*) dependence is hysteretic, with saturation in magnetic fields above 0.15 T.

The magnitude of the magnetoresistance (MR) in sample S1 is small. At temperatures below 70 K, the negative magnetoresistance (NMR) of less than 0.05 % at 0.3 T was observed. The Curie temperature of the InAsSb:Fe layer of sample S1 can be estimated as approx. 40 K, while the Fe concentration is about 15 at. %.

Figure 12a presents the *R*_H_(*B*) dependences in the temperature range of 30–300 K for sample S2. In contrast to sample S1, the *R*_H_(*B*) dependences for sample S2 are dominated by the anomalous Hall effect up to at least 300 K. The sign of the *R*_H_(*B*) curves corresponds to the *p*-type conductivity, while according to the sign of the Seebeck coefficient, the InAsSb:Fe layer of sample S2 (as well as sample S1) exhibits *n*-type conductivity. Note that the difference in the sign of the ordinary and anomalous Hall effect was also detected for structure S1 (Figure 11). This is also characteristic of the (In,Fe)As [3] and (In,Fe)Sb layers [6].

Figure 12b shows the magnetoresistance curves *MR*(*B*) for structure S2 at different temperatures (with the magnetic field applied perpendicularly to the sample plane). A pronounced NMR, hysteretic at temperatures below 200 K, was observed. Thus, in contrast to sample S1, the Curie temperature of the InAsSb:Fe layer of sample S2 is at least 200 K. For sample S2, the *R*_H_(*B*) and *MR*(*B*) dependences are similar to those for the PLD-grown (In,Fe)Sb layers [6].

The results obtained for structures S1 and S2 allow us to conclude that the transport and magnetic properties of the InAsSb:Fe layers strongly depend on the ratio of the Sb and As concentrations. The increase in the Sb concentration enhances the ferromagnetic properties (the magnitudes of the AHE, NMR and Curie temperature) of the InAsSb:Fe magnetic semiconductor and (as expected) brings it closer to the properties of the (In,Fe)Sb magnetic semiconductor. The nature of the ferromagnetic exchange interaction in the III–V:Fe magnetic semiconductors is currently the subject of research, and no undisputed opinion can be pointed out. There is the assumption that the ferromagnetism in (In,Fe)As is related to the mechanism of indirect ferromagnetic exchange via charge carriers [3]. At the same time, the magnetic properties of (In,Fe)Sb do not depend on the concentration and type of the majority charge carriers [14] and are probably associated with some mechanism of ferromagnetic exchange consisting of the overlapping of the wave functions of Fe atoms via intermediate atoms (i.e., some kind of superexchange mechanism). The observed enhancement of the magnetic properties of InAsSb:Fe with increasing Sb content indicates that the indirect ferromagnetic exchange interaction between Fe atoms via Sb atoms is stronger than that via As or In atoms.

Figure 13a shows the *R*_H_(*B*) dependences in the temperature range of 10–300 K for the InGaSb:Fe layer of sample S3. The anomalous Hall effect dominates the *R*_H_(*B*) dependences up to at least 300 K. In particular, the *R*_H_(*B*) dependences agree with the magnetic field dependences of the Hall resistivity for the MBE-grown (Ga,Fe)Sb layers [15].

The sign of the *R*_H_(*B*) curves corresponds to the *p*-type conductivity, and the sign of the Seebeck coefficient also reveals the *p*-type conductivity of the InGaSb:Fe layer in sample S3. The *p*-type conductivity is characteristic of heavily Fe-doped GaSb layers, which is related to the native acceptor point defects in the GaSb matrix. In the InGaSb:Fe layer of sample S3, the concentration of In atoms does not exceed 10 at. % (Table 2, Figure 6 and Figure 9). Therefore, the InGaSb:Fe layer should be closer in its properties to the (Ga,Fe)Sb compound. For structure S3, in addition to the anomalous Hall effect, a pronounced NMR at temperatures up to 300 K was also observed (Figure 13b). From the Arrott plots of the *R*_H_(*B*) dependencies, it was concluded that the Curie temperature for the InGaSb:Fe layer of sample S3 is about 300 K.

The presented results allow us to conclude that pulsed laser deposition makes it possible to grow high-quality multilayer epitaxial heterostructures based on the various multi-component layers of III–V semiconductors heavily doped with Fe.

## 4. Conclusions

In summary, using the pulsed laser sputtering of solid targets in a vacuum, it is possible to form three-layer epitaxial heterostructures containing layers of various III–V:Fe multi-component magnetic semiconductors, such as InAsSb:Fe and InGaSb:Fe, with a Fe concentration in the range of 13–24 at. %. It has been found that the diffusion of Ga atoms from the GaAs regions leads to the appearance of up to 20 at. % of gallium in layers with the nominal composition InAsSb:Fe, i.e., to the formation of InGaAsSb:Fe layers. It has been established that the ferromagnetic properties (the magnitude of the anomalous Hall effect, negative magnetoresistance and Curie temperature) of the InAsSb:Fe magnetic semiconductor are enhanced with an increasing Sb:As ratio. Therefore, we conclude that the indirect ferromagnetic exchange interaction between Fe atoms occurs predominantly via Sb atoms.

## Figures and Tables

**Figure 1 nanomaterials-13-02435-f001:**
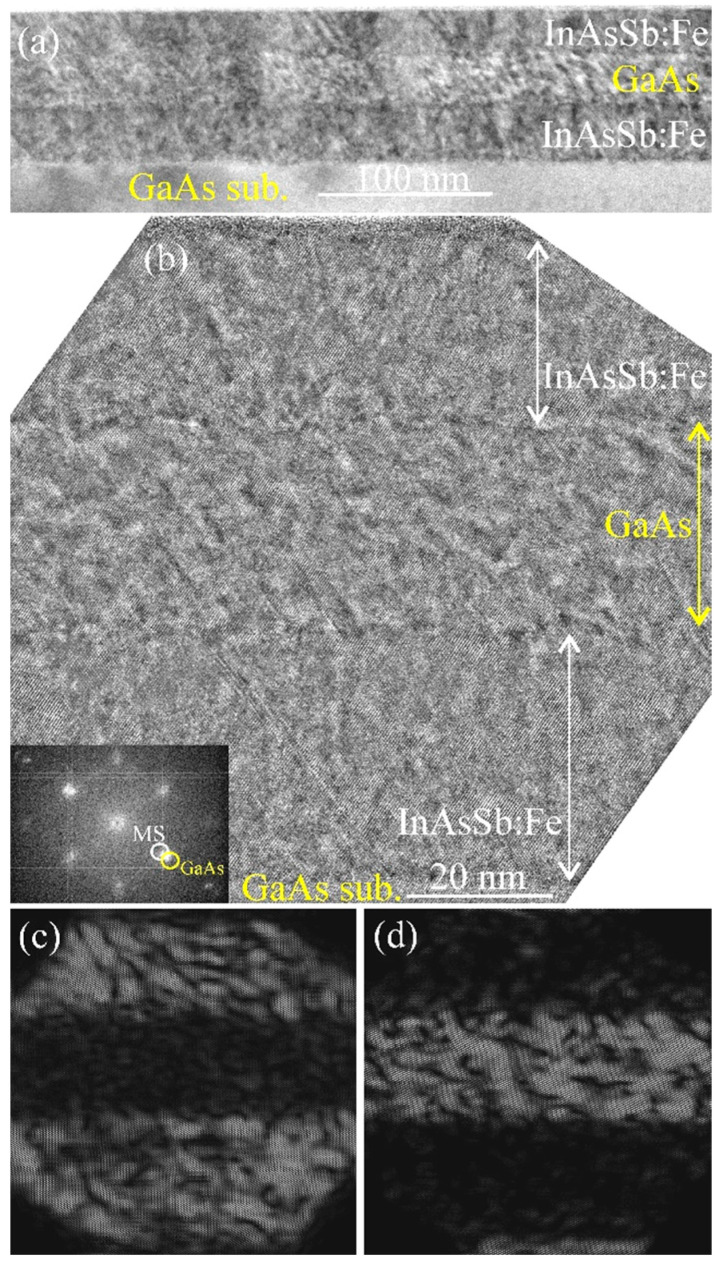
Cross-section TEM images of the InAsSb:Fe/GaAs/InAsSb:Fe/GaAs(sub.) sample M1. (**a**) Overview TEM image. (**b**) HRTEM image. The inset shows the respective FFT diffraction pattern. (**c**) Inverse FFT image of the Fe-doped regions obtained from MS spots. (**d**) Inverse FFT image of the GaAs regions obtained from GaAs spots.

**Figure 2 nanomaterials-13-02435-f002:**
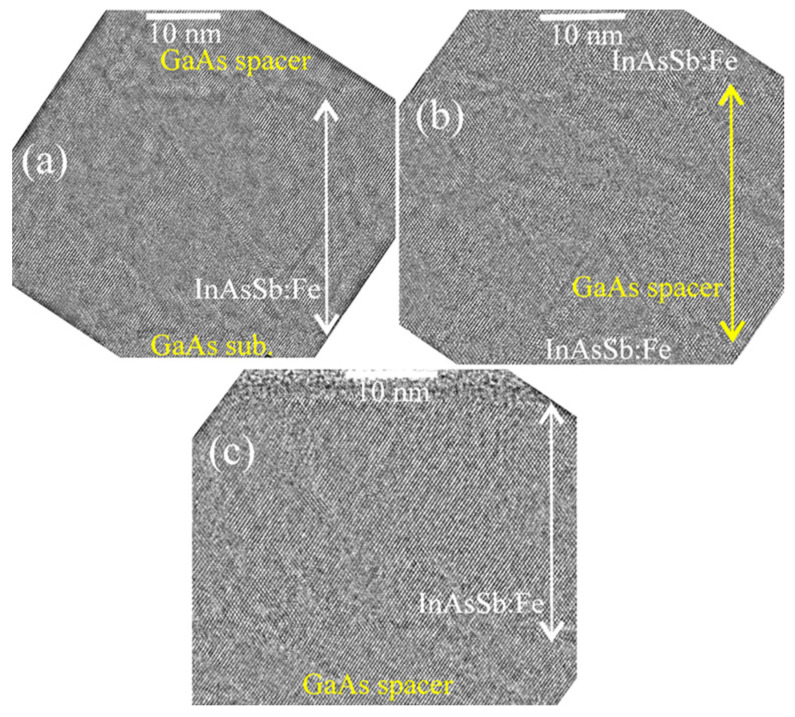
Cross-section HRTEM images of the layers of sample M1. (**a**) HRTEM image of the lower InAsSb:Fe layer. (**b**) HRTEM image of the GaAs spacer. (**c**) HRTEM image of the upper InAsSb:Fe layer.

**Figure 3 nanomaterials-13-02435-f003:**
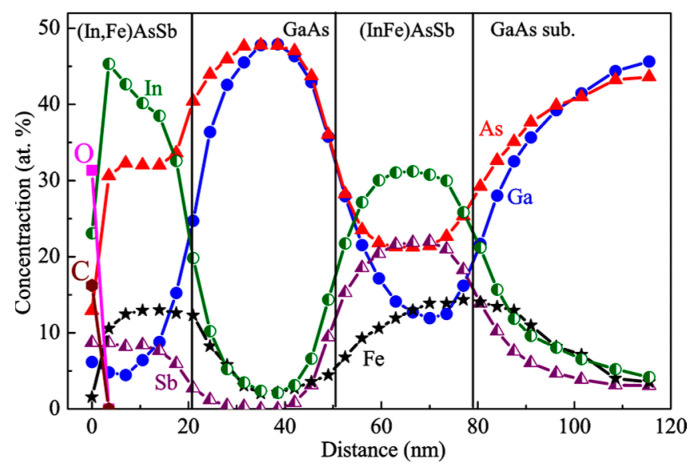
XPS depth distribution profiles of C, O, In, Ga, As, Sb and Fe atoms from the surface to the substrate for sample M1.

**Figure 4 nanomaterials-13-02435-f004:**
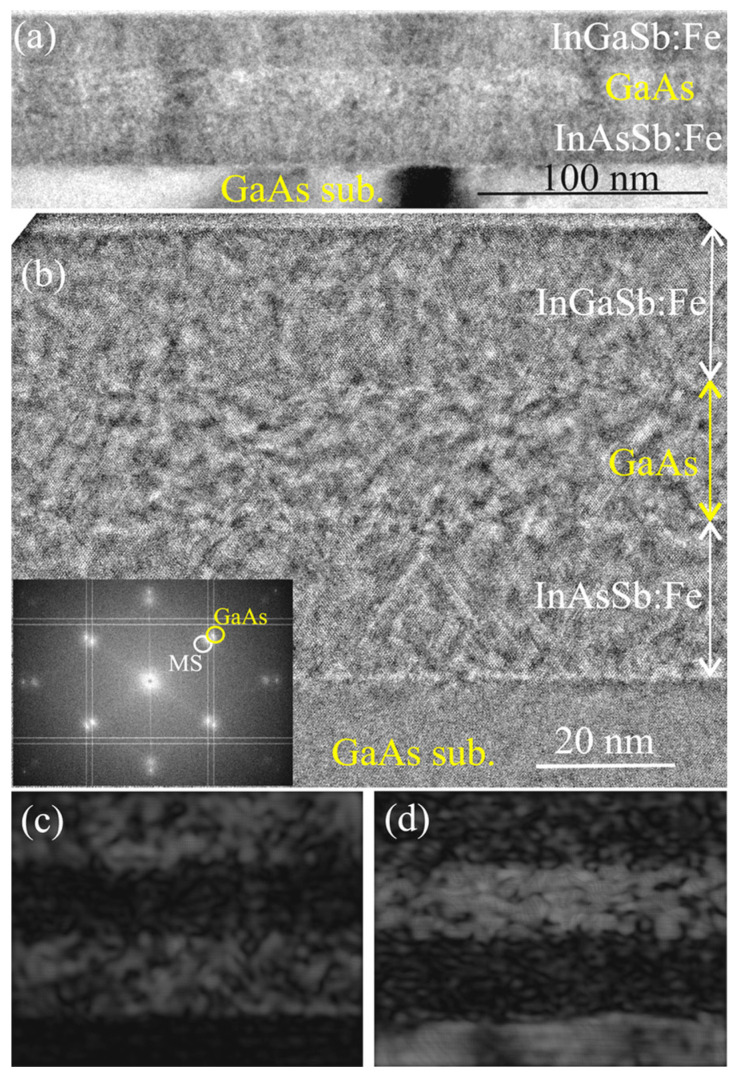
Cross-section TEM images of the InGaSb:Fe/GaAs/InAsSb:Fe/GaAs(sub.) sample M2. (**a**) Overview TEM image. (**b**) HRTEM image. The inset shows the respective FFT diffraction pattern. (**c**) Inverse FFT image of the Fe-doped regions obtained from MS spots. (**d**) Inverse FFT image of the GaAs regions obtained from GaAs spots.

**Figure 5 nanomaterials-13-02435-f005:**
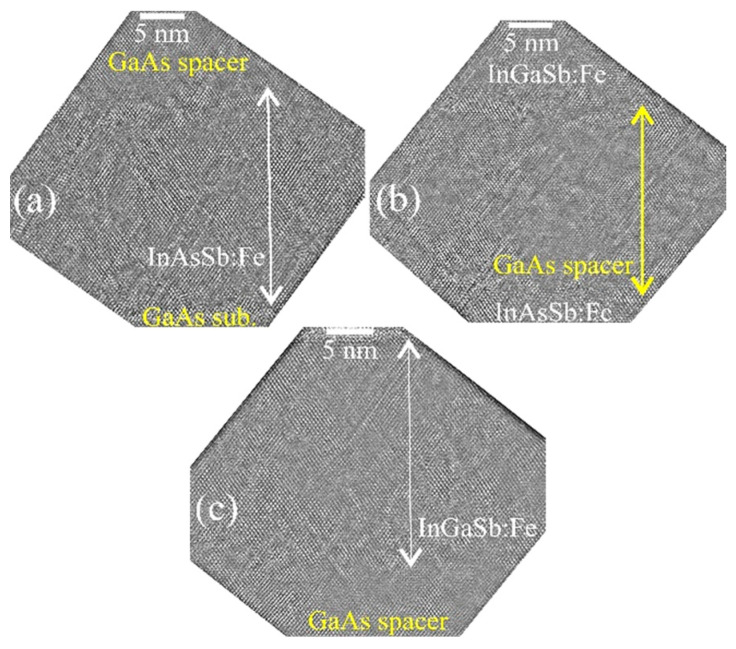
Cross-section HRTEM images of each layer of sample M2. (**a**) HRTEM image of the lower InAsSb:Fe layer. (**b**) HRTEM image of the GaAs spacer. (**c**) HRTEM image of the upper InGaSb:Fe layer.

**Figure 6 nanomaterials-13-02435-f006:**
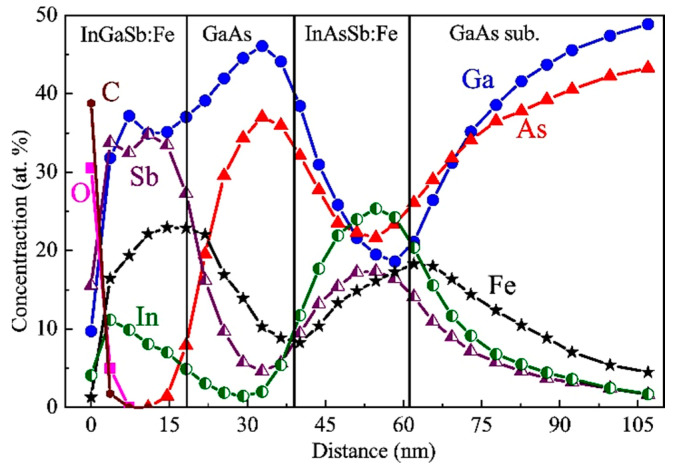
XPS depth distribution profiles of C, O, In, Ga, As, Sb and Fe atoms from the surface to the substrate of sample M2.

**Figure 7 nanomaterials-13-02435-f007:**
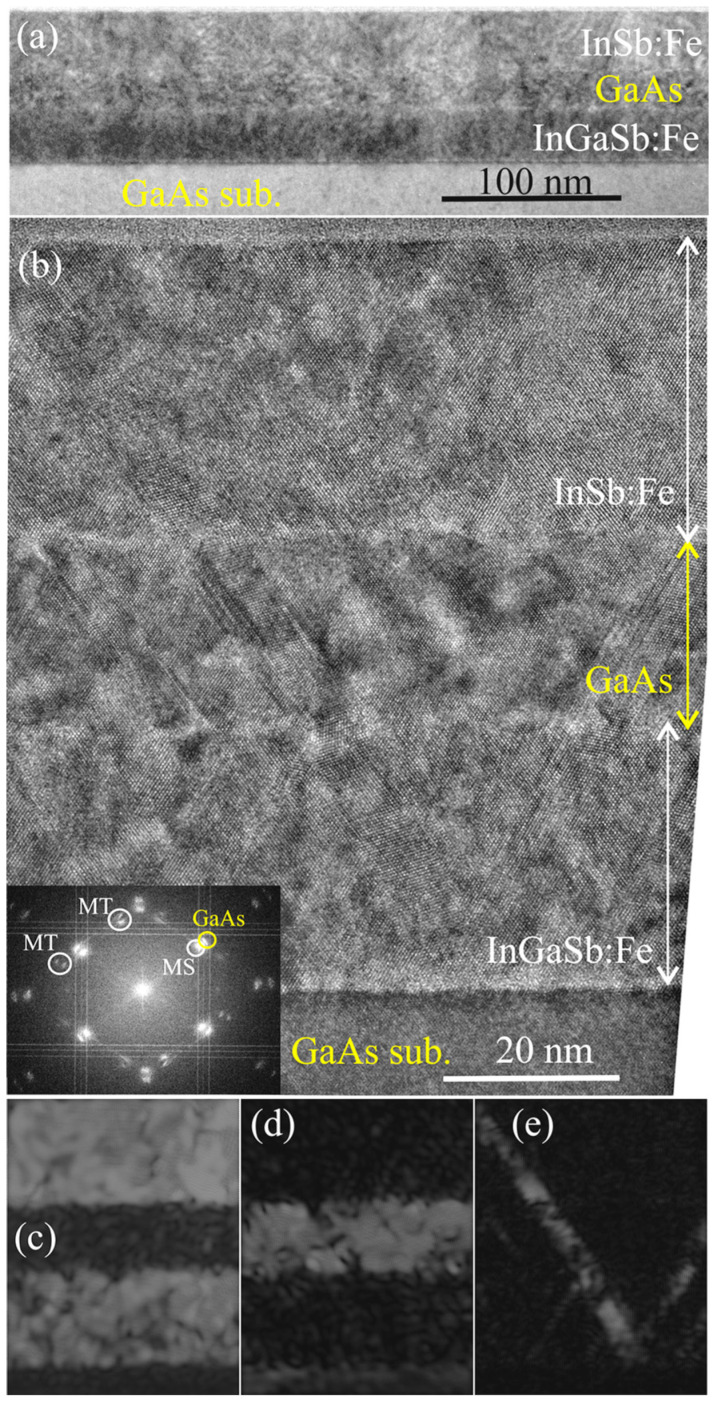
Cross-section TEM images of the InSb:Fe/GaAs/InGaSb:Fe/GaAs (sub.) structure of M3. (**a**) Overview TEM image. (**b**) HRTEM image. The inset shows the respective FFT diffraction pattern. (**c**) Inverse FFT image of the Fe-doped regions obtained from MS spots. (**d**) Inverse FFT image of the GaAs regions obtained from GaAs spots. (**e**) Inverse FFT image obtained from additional spots.

**Figure 8 nanomaterials-13-02435-f008:**
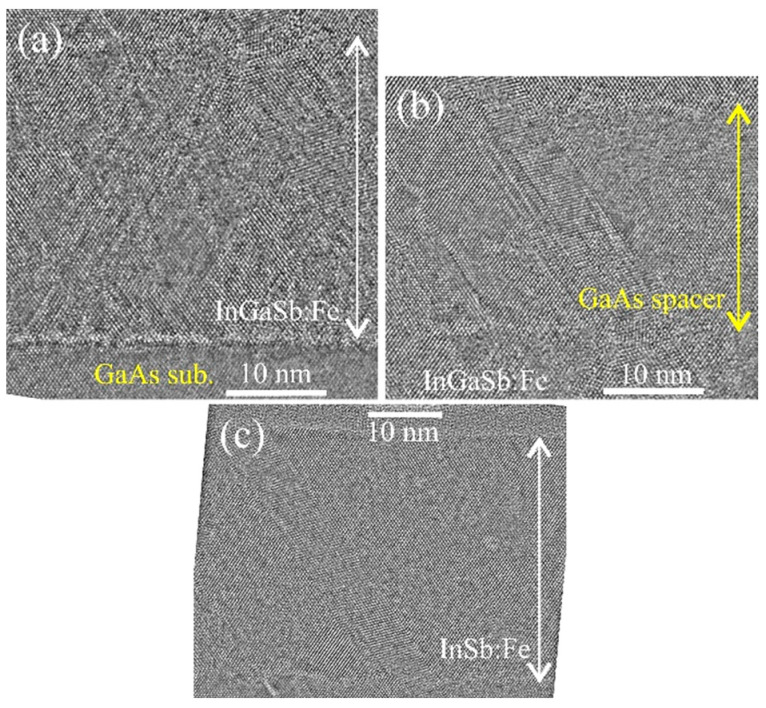
Cross-section HRTEM images of each layer of sample M3. (**a**) HRTEM image of the lower InGaSb:Fe layer. (**b**) HRTEM image of the GaAs spacer. (**c**) HRTEM image of the upper InSb:Fe layer.

**Figure 9 nanomaterials-13-02435-f009:**
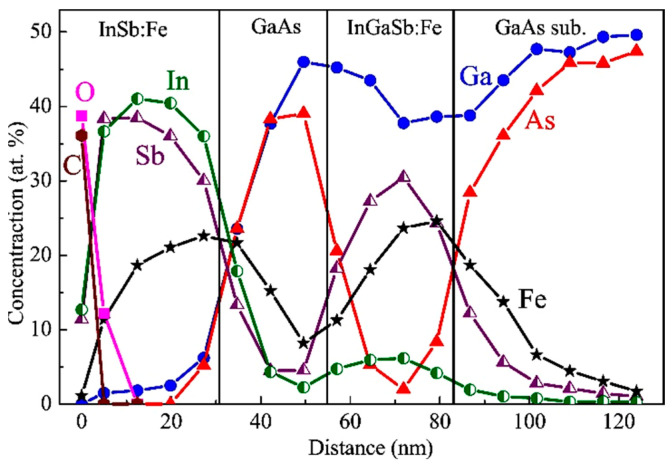
XPS depth distribution profiles of C, O, In, Ga, As, Sb and Fe atoms from surface to substrate for structure M3.

**Figure 10 nanomaterials-13-02435-f010:**
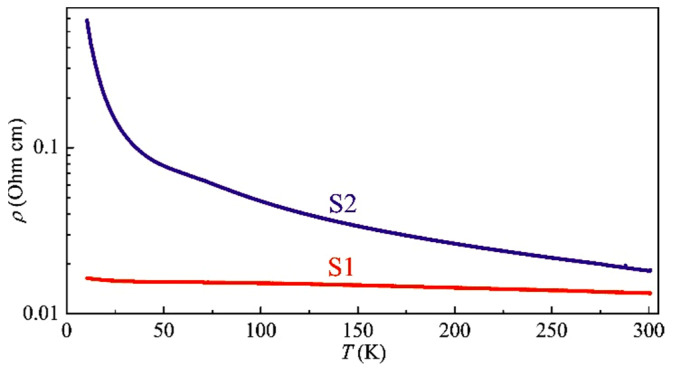
Temperature dependences of the resistivity for samples S1 and S2.

**Figure 11 nanomaterials-13-02435-f011:**
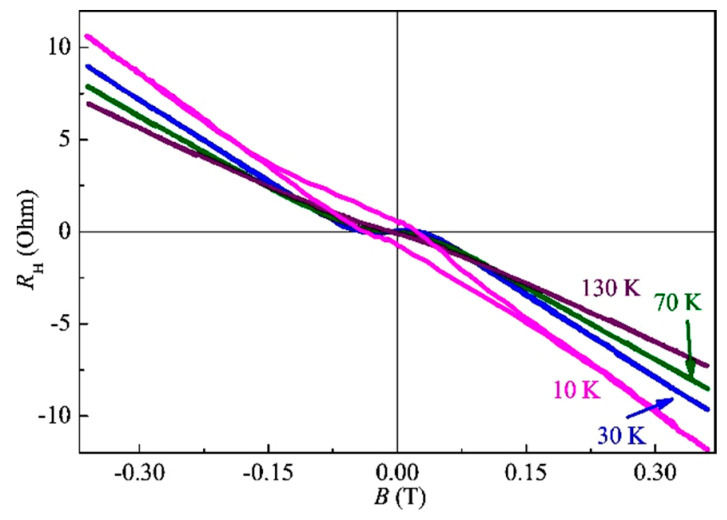
*R*_H_(*B*) dependences at various temperatures for sample S1.

**Figure 12 nanomaterials-13-02435-f012:**
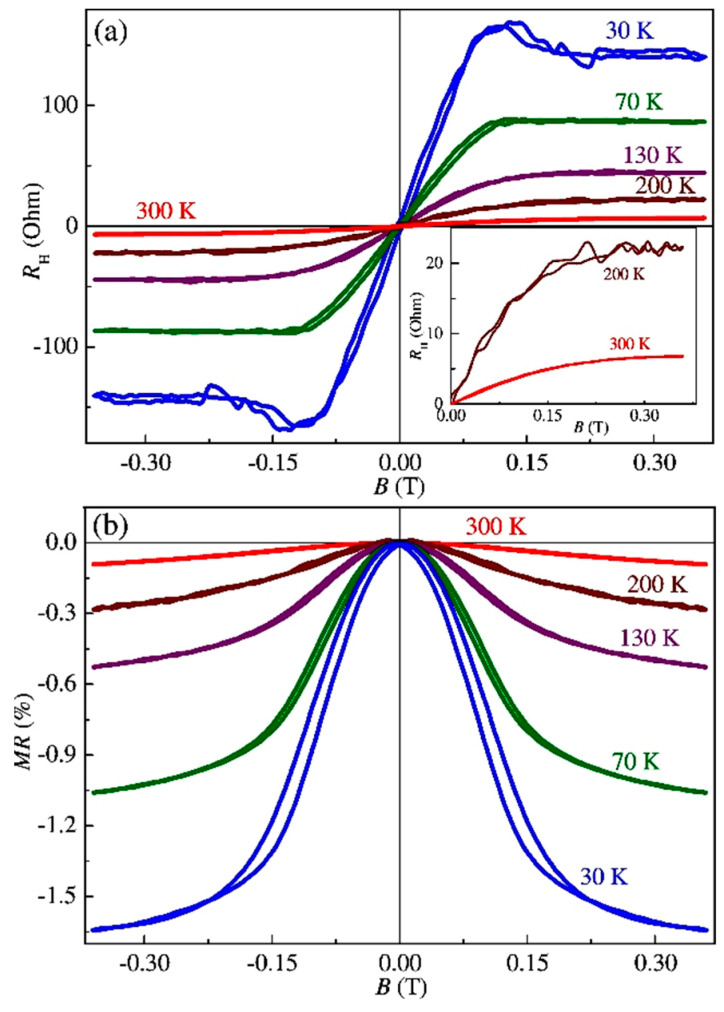
*R*_H_(*B*) dependences (**a**) and magnetoresistance (**b**) at various temperatures for sample S2.

**Figure 13 nanomaterials-13-02435-f013:**
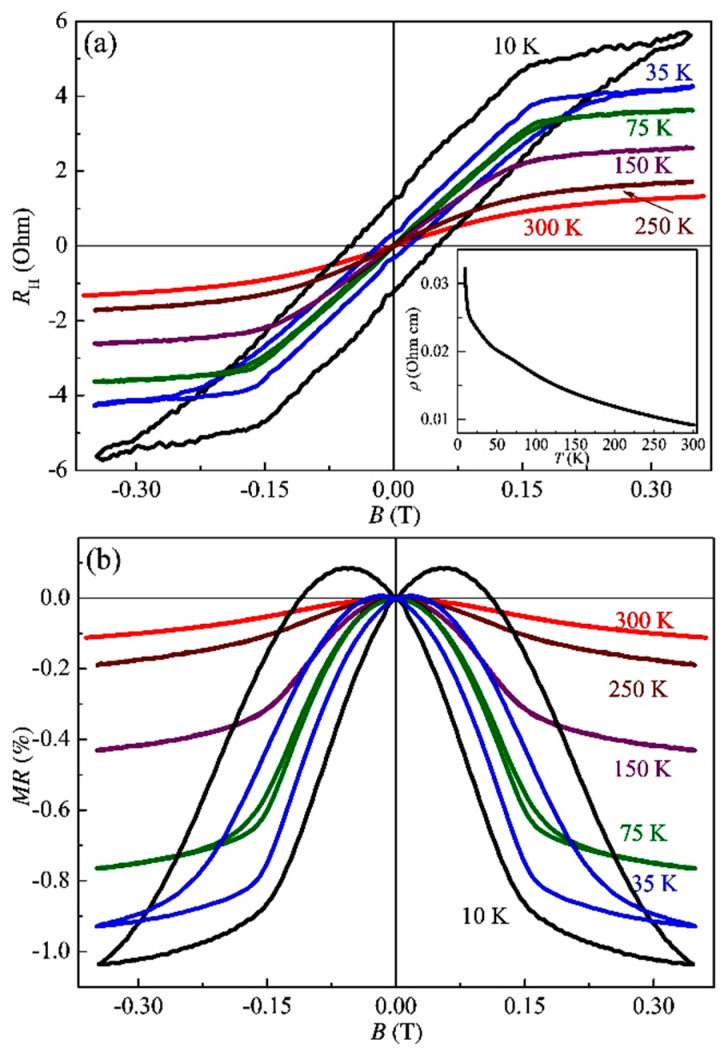
*R*_H_(*B*) dependences (**a**) and magnetoresistance (**b**) at various temperatures for sample S3.

**Table 1 nanomaterials-13-02435-t001:** The investigated three-layer samples and their nominal compositions.

Sample	Structure
M1	(InAs_0.8_Sb_0.2_)_0.83_Fe_0.17_/GaAs/(InAs_0.5_Sb_0.5_)_0.83_Fe_0.17_/*i*-GaAs substrate
M2	(In_0.2_Ga_0.8_Sb)_0.8_Fe_0.2_/GaAs/(InAs_0.5_Sb_0.5_)_0.83_Fe_0.17_/*i*-GaAs substrate
M3	(InSb)_0.8_Fe_0.2_/GaAs/(In_0.2_Ga_0.8_Sb)_0.8_Fe_0.2_/*i*-GaAs substrate

**Table 2 nanomaterials-13-02435-t002:** The investigated single-layer samples and their nominal compositions.

Sample	Structure
S1	(InAs_0.8_Sb_0.2_)_0.83_Fe_0.17_/*i*-GaAs
S2	(InAs_0.5_Sb_0.5_)_0.83_Fe_0.17_/*i*-GaAs
S3	(In_0.2_Ga_0.8_Sb)_0.8_Fe_0.2_/*i*-GaAs

## Data Availability

Not applicable.

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
