# Peer review of "Multilayer Epitaxial Heterostructures with Multi-Component III–V:Fe Magnetic Semiconductors"

_nanomaterials, 2023, doi:10.3390/nano13172435_

Round 1

Reviewer 1 Report

The manuscript, entitled "Multilayer epitaxial heterostructures with multi-component 2 III–V:Fe magnetic semiconductors," presents an experimental investigation of trilayer heterostructures consisting of InAsSb:Fe, InGaSb:Fe and InSb:Fe layers with Fe concentrations up to 24 at. % separated by GaAs spacer grown on (001) i-GaAs substrates. The manuscript provides a comprehensive analysis of the film's structure and morphology through the use of transmission electron microscopy. Additionally, the trilayers were characterized through magnetotransport analysis. While the manuscript showcases several intriguing findings, it requires improvement in terms of its English language. Overall, I find this research to be relevant and interesting to the scientific community. However, before making any decision, I would like to share some comments that the authors should consider addressing:

1) Have the transverse magneto-responses been antisymmetrized with respect to the magnetic field to obtain the purely field-antisymmetric Hall?

2) Why do the authors use the term "Seebeck coefficient" when they don't present thermoelectric measurements? They are mixing the Seebeck coefficient with the Hall coefficient.

3) In order to refer to epitaxial growth, it is necessary that only one set of planes is present in the direction perpendicular to the surface. As a result, X-ray diffraction (XRD) measurements are required to corroborate their assertion, as data from transmission electron microscopy (TEM) alone is insufficient.

4) When examining Figure 12, it is noticeable that the blue AHE gathered at 30 K appears to have some bumps near 0.15 T. It would be helpful to understand the cause of these bumps. Could they be connected to the band structure or the two-channel conduction? It is suggested that the authors provide further elaboration on this matter in their discussion.

5) In Figure 13, at a temperature of 10 K, the AHE loop displays significant coercivity. Is there a magnetic phase transition occurring around this temperature that causes it to behave differently than at higher temperatures? Further discussion is needed regarding this aspect.

The English language requires improvement. Authors should consider utilizing an English editing service or seeking assistance from a native English-speaking colleague

Author Response

See the file in attachment

Round 2

Reviewer 1 Report

After careful consideration, I am satisfied with the authors' responses to my concerns. As a result, I highly recommend their manuscript for publication in Nanomaterials. 

Reviewer 2 Report

The revision can be accpeted.